# Influence of Thermal Processing on the Bioactive, Antioxidant, and Physicochemical Properties of Conventional and Organic Agriculture Black Garlic (*Allium sativum* L.)

**Katarzyna Najman** *, **Anna Sadowska** and **Ewelina Hallmann**

Department of Functional and Organic Food, Institute of Human Nutrition Sciences, Warsaw University of Life Sciences, Nowoursynowska 159c, 02-776 Warsaw, Poland; anna_sadowska@sggw.edu.pl (A.S.); ewelina_hallmann@sggw.edu.pl (E.H.)
* Correspondence: katarzyna_najman@sggw.edu.pl; Tel.: +48-22-59-370-55

**Abstract:** This study examines the effect of fermentation processes (70 °C temperature; 80% humidity; 45 days) on the content of bioactive compounds (total polyphenols, selected phenolic acids, and flavonoids), antioxidant activity and physicochemical properties of conventional and organic garlic (*Allium sativum* L.). Raw garlic from conventional cultivation (GR) showed significantly lower content of phenolic components and antioxidant activity compared to raw garlic from organic cultivation (GREcol) (by 22.8% and 25.1%, respectively). The fermentation processes of the garlic from both the conventional (BG) and organic (BGEcol) cultivation significantly increased the content of total polyphenols (more than two times), phenolic acids (more than 1.5 times) and flavonoids (1.5 and 1.3 times, respectively). The garlic fermentation process also significantly increased antioxidant potential: two times for BG and three times for BGEcol. The content of bioactive compounds (total polyphenols, phenolic acids, and flavonoids) was significantly ($p < 0.05$) higher in organic black garlic (BGEcol) compared to conventional black garlic (BG). Heat treatment leading to changes in both the physicochemical, organoleptic and health-promoting properties allowed to obtain a new product gaining in sensory attractiveness and enabling a wide range of applications.

**Keywords:** black garlic; *Allium*; bioactive compounds; polyphenols; antioxidant activity; organic food

## 1. Introduction

Medical and culinary use of garlic (*Allium sativum* L.) has a long history—its great value has been known since antiquity. Garlic is the earliest documented medicinal plant [1]. In traditional medicine, it was used in the treatment of various diseases, including ear diseases, deafness, leprosy, severe diarrhea, constipation, parasitic infections, fever, abdominal pain, and infections. Its prophylactic and therapeutic effects have been confirmed by numerous contemporary studies [2,3]. This plant has a diaphoretic, expectorant, antispasmodic, bacteriostatic, antiviral, and anthelmintic effect. It is commonly used in the treatment of colds, bronchitis and upper respiratory tract inflammation, recurrent infections, and flu [4–6]. Modern medicine turns to this plant again, mainly due to its high content and great variety of bioactive compounds that have a positive effect on the body. The health-promoting effect of garlic are related to its high content of vitamins (C and B), mineral compounds (selenium, calcium, potassium, and copper), and polyphenols, which, together with organic sulfur compounds, contribute to the high antioxidant potential and medical properties of garlic [7–10].

Garlic is a plant widespread across the world and used in international cuisine due to its unique taste. The clear, characteristic, sharp taste of garlic is created in complex biochemical reactions.

The main compounds determining both the flavor and health-promoting properties are organic sulfur compounds [11–13]. These compounds are credited with the wide range of medicinal properties attributed to garlic, such as anti-cancer, anti-diabetic, anti-inflammatory, anti-microbial, anti-oxidant, immunomodulating, cardioprotective, and anti-atherosclerotic properties [13–16]. Intensive research in both human and experimental animal studies (including our own) has shown that garlic has a beneficial effect on the level of blood lipids, improving, e.g., the lipid profile by lowering blood cholesterol and triacylglycerol levels [17–20]. Numerous experiments confirm the hypotensive, anti-aggregating, anti-inflammatory, and antioxidant effects [2,11,15,21–23]. Intensive studies on the antioxidant properties of garlic, both of its active substances and various preparations, forms and processing products, provide knowledge about the capacity of garlic to scavenge free radicals and interrupt free radical reactions, and to strengthen the natural enzymatic antioxidant defense of the body, confirming the legitimacy of its use in the prevention of metabolic diseases, including atherosclerotic diseases but also in the prevention and alleviation of the course of neoplastic diseases [21,22,24–30]. Due to its multidirectional beneficial effects, garlic is nowadays recommended as a dietary supplement [31].

The health-promoting, dietary, and nutritional properties of garlic make it widely used in all cuisines of the world. As an addition in the form of a spice to dishes (fresh, macerated, as old garlic extract, as oil, in flakes, as powder, dried, and freeze-dried), it provides them with a unique, characteristic, sharp taste and aroma, whose precursor are the organosulfur compounds present in garlic, which are produced during crushing and processing [32–34]. Sulfur-containing compounds present in fresh garlic include, e.g., S-allyl-L-cysteine sulfoxide (alliin) and γ-glutamyl-S-allyl-L-cysteine, under the influence of allinase are hydrolyzed, e.g., to the sulfides (DAS), disulfides (DADS) and trisulfides (DTS) of diallyl and ajoenes [35–37] and other organosulfur metabolites such as S-allylcysteine (SAC) contributing to the numerous health benefits of garlic, including its antioxidant properties [26,29,36,38]. Therefore, consumption of fresh garlic is necessarily associated with the accompanying characteristic, sharp aroma, which is not always desired by the consumers.

Technological processes, depending on type and intensity, affect the properties, activity, and bioavailability of compounds contained in fruits and vegetables, bringing about changes in their concentration, activity, and structure [39]. The most common methods of garlic processing include heat treatment (cooking, baking, frying, or microwaving), which may change the properties of the dietary fiber, its quantity and proportions of its fractions, cause loss of vitamins and other biologically active compounds, and reduce the content of natural antioxidants [19,33,34]. Thermal treatment, depending on its type and duration, may modify the composition and mutual proportions, as well as the bioavailability of bioactive compounds, e.g., organosulfuric or polyphenolic compounds, thus influencing the physiological effects of garlic consumption in the body [33,34,39].

Therefore, new methods of garlic processing, production technologies, and industrial processes are constantly being searched for, which, on one hand, will allow to eliminate the unpleasant and irritating smell, and on other will improve the palatability and organoleptic characteristics and help obtain a better quality product, securing or even enhancing the beneficial properties of this nutritious plant. An example of such a process is the garlic fermentation process leading to the production of black garlic—a product characterized by completely different physicochemical, sensory, culinary, nutritional, dietary, and health-promoting properties than raw garlic [36,37,40–44]. In recent years, black garlic has been gaining ever more interest from both consumers and scientists, and thanks to its incomparably better health-promoting properties than those of other processed products and raw garlic, it is now considered the healthiest food product of natural origin and classified as a functional food of the "Superfood" kind.

The process of producing black garlic, known from antiquity, comes from Asia and has been part of Korean, Japanese, and Thai cuisine for centuries. Over a dozen years ago was introduced to Europe, where it enjoys a constantly growing consumer interest. Black garlic is produced by long-lasting fermentation processes of whole heads or cloves kept under controlled conditions of high temperature (60–90 °C) and humidity (70–90%) for a period of several to over a many weeks

(usually 30–90 days). This process takes place without the participation of microorganisms (bacteria and fungi), and without the addition of preservatives, but only under the influence of enzymes naturally occurring in the raw material, as a result of natural reactions and biochemical changes of the compounds present in garlic [36,37,40–45]. During this long-term heat treatment, garlic undergoes enzymatic and non-enzymatic browning reactions, e.g., oxidation, caramelization and Maillard reactions, which radically change the physicochemical, organoleptic, sensory, and bioactive properties of garlic [46,47]. As a result of fermentation processes, unstable organosulfur compounds present in raw garlic are transformed into stable, water-soluble substances, the most important of which is S-allylcysteine, displaying very high antioxidant activity. For this reason, black garlic exhibits many times higher antioxidant potential than raw garlic [44,46,47].

As the temperature of the fermentation process increases, so does the intensity of the production of browning reaction products, including Amadori and Heyns compounds, which are important intermediates in the Maillard reaction that can further generate various thermal treatment products [40,46,47]. Since, as a result of the reaction of glucose or fructose with amino acids, changes in the sugar content occur, the fructose content increases significantly, thus leading to the characteristic sweetness of black garlic. As a result, it loses its sharp, irritating, characteristic garlic flavor and aroma. With the increase in the content of browning reaction products, the pH drops, e.g., as a result of the synthesis of organic acids resulting from the oxidation of aldehyde groups in aldoses. The enzymatic and non-enzymatic browning reactions are responsible for the change of garlic's color from white, through caramel, brown, to black, as well as for the appearance of sweet and sour taste of black garlic, with a hint of dried plum and apricot, vanilla or licorice [44,47]. In addition, the texture changes to a more rubbery or "jelly-like", and the consistency becomes soft, delicate, resembling cream cheese. As a result of these long-lasting processes, a completely new product is created, gaining in sensory attractiveness and having wide possibilities of its application.

In addition to the processing and various technological operations, under the influence of which the properties of both garlic and other fruit or vegetable raw materials change dramatically [48], the chemical composition and content of bioactive and organosulfuric substances and sugars in garlic are also influenced by other factors, e.g., genetic (species and variety) or environmental [49–52]. Climatic conditions, the cultivation system (conventional/ecological), sulfur fertilization, harvesting, and storage conditions have an equally important impact on the quality of raw materials, and thus on the quality of the products obtained from them [12,53–56]. There are many studies in the literature confirming that fruit and vegetable raw materials from organic farming are characterized by a higher content of bioactive compounds, such as polyphenols, flavonoids, carotenoids, sugars, and others, and show a higher antioxidant potential, and thus a higher nutritional, dietary, and health-promoting value [57–59].

In the available literature, there are studies on the properties of black garlic, the influence of the fermentation process conditions (time, humidity, and temperature) on the composition and proportions of the biologically active compounds, or on the mechanisms of enzymatic and non-enzymatic browning reactions occurring in garlic under various aging conditions, but it was not found in available literature the comparative data concerning the content of bioactive compounds, as well as physicochemical properties in black garlic from conventional and organic cultivation. Therefore, the aim of this study was to compare the effect of fermentation processes (aging of garlic), conducted in controlled conditions of high humidity (80%) and temperature (70 °C) during 45 days, on the content of bioactive compounds (including total polyphenols, selected phenolic acids, and flavonoids), antioxidant activity and physicochemical properties of garlic from conventional and organic cultivation. Such research is important not only for scientists, but also for food producers, including organic food producers or for food technologists who design new, functional products with a wide range of interesting physicochemical, technological, and health-promoting properties. The results of the comparative analysis can also provide valuable tips for consumers, whose nutritional awareness, interest in organic food, and care for health and the quality of meals are constantly growing. The modern food and

processing industry must meet not only the requirements as to the quality and health standards of food products, but also meet the high demands of the consumers looking for food with high nutritional, dietary and health-promoting values. Therefore, comparative research on physicochemical, technological, bioactive, and health-promoting properties is important for many industries and the food market.

## 2. Materials and Methods

### 2.1. Materials

The study used garlic (*Allium sativum* L.), of the "Harnaś" cultivar, grown in 2019 year in the conventional and ecological cultivation system (from farms located in the Warmian-Masurian Voivodeship, Lubawa, Poland). Before starting the thermal treatment processes, the fresh material was cleaned and stored in a refrigerator at 10 °C and 85% relative air humidity for 24 h. Garlic from two cultivation systems (conventional and organic) was divided into two parts, one of which was intended for physicochemical studies of fresh material (GR and GREcol), and the other for the production of fermented black garlic (BG and BGEcol). The process of thermal treatment (fermentation) of the whole garlic heads was carried out in a climate chamber (Humidity Chamber HCP70, Memmert GmbH & Co. KG, Schwabach, Germany) with a controlled temperature (70 °C) and humidity (80%) for 45 days. Fresh test material (GR and GREcol) and fermented black garlic (BG and BGEcol) were peeled, cut into cloves and homogenized for 1 min (10,000 rpm), (IKA T18 Basic ULTRA-TURRAX®, IKA® Poland Ltd., Warsaw, Poland), packed into sterile plastic Falcone tubes and intended for testing.

### 2.2. Methods

#### 2.2.1. Preparation of Water Extracts for Physicochemical Analyses

To prepare the extracts 1.0 g of homogenized garlic was weighed (with an accuracy up to 0.001 g) on an analytical balance (AS 220/X, Radwag, Radom, Poland) into sterile plastic Falcone tubes with screw caps (50 mL capacity), and 30 mL of distilled water was added. It was shaken on a Vortex shaker (Wizard Advanced IR Vortex Mixer, VELP Scientifica Srl, Usmate, Italy) for 60 s (2000 rpm) for thorough mixing. Then it was incubated in a shaking incubator (IKA KS 4000i Control, IKA® Poland Ltd., Warsaw, Poland) for 60 min (40 °C, 200 rpm). After the incubation, the sample was re-shaken on a Vortex shaker for 60 s for thorough mixing and then centrifuged in a refrigerated centrifuge (MPW-380 R, MPW Med. Instruments, Warsaw, Poland) for 15 min (4 °C, 10,000 rpm), and the obtained supernatant was designated for the determination of the antioxidant activity and the total content of total polyphenols.

#### 2.2.2. Preparation of Methanol Extracts to Separate Polyphenolic Compounds

To prepare the extracts 0.5 g of homogenized garlic was weighed (with an accuracy up to 0.001 g) on an analytical balance (AS 220/X, Radwag, Radom, Poland) into sterile plastic tubes (10 mL capacity) and 5.0 mL of 80% methanol (Sigma-Aldrich, Poznań, Poland) was added. It was shaken on a Vortex shaker (Wizard Advanced IR Vortex Mixer, VELP Scientifica Srl, Usmate, Italy) for 60 s (2000 rpm) for thorough mixing and then incubated in an ultrasonic bath (Bandelin Sonorex RK 255, BANDELIN Electronic GmbH & Co. KG, Berlin, Germany) for 10 min (35 kHz, 30 °C). After the incubation, the samples were centrifuged in a centrifuge (MPW-380 R, MPW Med. Instruments, Warsaw, Poland) for 15 min (2 °C, 6000 rpm). The obtained supernatant (1.0 mL) was collected into special chromatographic vials and subjected to chromatographic analysis.

#### 2.2.3. Antioxidant Activity

The antioxidant activity in the aqueous extracts of the research material was determined using the cation radical $ABTS^{+\bullet}$ (2,2′-azino-bis (3-ethylbenzothiazoline-6-sulfonic) acid) ($ABTS^{+\bullet}$, Sigma-Aldrich,

Poznań, Poland) by the method of Re et al. [60]. A defined amount of solution of the examined extracts, predetermined by the dilution scheme (0.5–1.5 mL) was measured into 10 mL glass test tubes, and then 3.0 mL of the ABTS$^{+\bullet}$ cation radical solution in PBS (Phosphate Buffer Solution, Sigma-Aldrich, Poznań, Poland) was added. The absorbance was measured after exactly 6 min of incubation at room temperature. Absorbance was measured at a wavelength of $\lambda = 734$ nm using a spectrophotometer (UV/Vis UV-6100A, Metash Instruments Co., Ltd., Shanghai, China). The results, after taking into account the dilution schemes used and calculation based on the calibration curve for Trolox (Sigma-Aldrich, Poznań, Poland) (y = −5.6153 + 0.7123, $R^2 = 0.9998$), were expressed as μM TEAC (Trolox Equivalent Antioxidant Capacity), i.e., the amount of μmoles of Trolox per 1 g of dry mass (d.m.). The determinations were performed in six independent replications.

### 2.2.4. The Total Content of Total Polyphenols

The total content of total polyphenols in the aqueous extracts of the test material was determined using the Folin–Ciocalteu (Sigma-Aldrich, Poznań, Poland) reagent by a modified method of Singleton and Rossi [61]. A defined amount of solution of the examined extracts, predetermined by the dilution scheme (1.0 mL) was collected into 50 mL graduated flasks, and then 2.5 mL of Folin–Ciocalteu reagent and 5.0 mL of sodium carbonate (Sigma-Aldrich, Poznań, Poland) with a concentration of 20% were added and made up to the mark with distilled water. The samples were incubated for 60 min at room temperature and protected from light. Absorbance was measured at a wavelength of $\lambda = 734$ nm using a spectrophotometer [UV/Vis UV-6100A, Metash Instruments Co., Ltd., Shanghai, China]. The results, after taking into account the applied dilution schemes and calculation based on the calibration curve for gallic acid (Sigma-Aldrich, Poznań, Poland) (y = 2.123 + 0.1327, $R^2 = 0.9992$), were expressed as mg GAE (Gallic Acid Equivalent), i.e., mg of gallic acid per 1 g of dry mass (d.m.). The determinations were performed in six independent replications.

### 2.2.5. Content and Separation of Polyphenolic Compounds (HPLC)

The content of selected phenolic compounds (phenolic acids, flavonoids) in garlic was determined by HPLC (High-Performance Liquid Chromatography), according to Hallmann et al. [62], using the Shimadzu HPLC kit (USA Manufacturing Inc., Waltham, MA, USA), consisting of two LC-20AD pumps, a CMB-20A system controller, a SIL-20AC autosampler, a SPD-20AV UV/VIS detector, and a CTD-20AC controller. Phenolic compounds were identified and separated on a Synergi Fusion-RP 80i (250 × 4.60 mm) chromatography column using a two phase flow gradient: acetonitrile/deionized water (55% and 10%) at pH 3.00. The analysis time was 38 min with a flow rate of 1.0 mL/min and detection at a wavelength of 250–370 nm. To identify the substances, standards substances from Fluka and Sigma-Aldrich of a purity of 99% were used. From the previously prepared standard solutions, 5 injections of standard phenolic acids and flavonoids were made, and then standard curves were prepared. The content of selected phenolic compounds (flavonoids and phenolic acids) in the tested material was calculated on the basis of the prepared standard curves for standard substances, and the results, taking into account the dilution schemes used, were expressed as mg/100 g of dry mass (d.m.). The determinations were made in three independent replications.

### 2.2.6. Determination of Dry Matter and Moisture Content of Garlic Samples

The dry mass of the garlic samples was determined gravimetrically according to the AOAC method [63] in a drying oven (SUP 200W, Wamed, Warsaw, Poland). The moisture content of the material was calculated from the difference in mass. The determination was made in three replications.

### 2.2.7. Determination of $a_w$ of the Garlic Samples

Water activity ($a_w$) was measured using the manual water activity meter with a temperature stabilizer AquaLab Water Activity Meter (Decagon Devices. Inc., Pullman, WA, USA). Measurements (in homogenates garlic cloves) were made in three independent replications.

### 2.2.8. Determination of the pH of Garlic Samples

Garlic pH was measured by the potentiometric method in a 1:5 mixture of homogenate with deionized water, using a laboratory pH meter probe (Elmetron CP-511, Elmetron G.P., Zabrze, Poland), at room temperature (20 °C). Measurements were made in three independent replications.

### 2.2.9. Statistical Analysis of the Results

The results are presented as the mean ± standard deviation (SD) of three to six independent replicates and analyzed using the statistical programme Statistica 13.0 (Tibco Software Inc., Palo Alto, CA, USA). One-way ANOVA with Duncan's test was performed. Differences at $p < 0.05$ were considered statistically significant. Principal component analysis (PCA) was used to assess the similarities and differences between the tested parameters of the garlic samples that were evaluated according to a covariance matrix.

## 3. Results and Discussion

Changes in the appearance of conventional (G) and organic (GEcol) garlic bulbs during the fermentation process (temperature 70 °C, 80% humidity, 45 days) are shown in Figure 1, while Table 1 presents a physicochemical properties of raw and black garlic from conventional and organic cultivation.

**Raw Garlic**　　　　　　　　　　　　　　　　　　　　　　**Black Garlic**

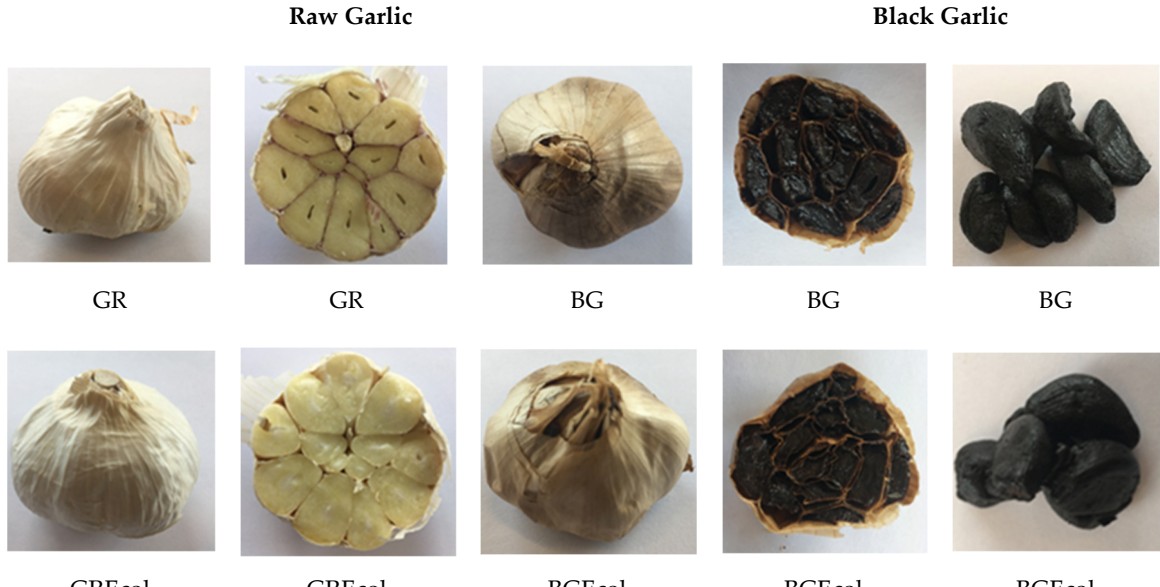

GR　　　　　　GR　　　　　　BG　　　　　　BG　　　　　　BG

GREcol　　　　GREcol　　　　BGEcol　　　　BGEcol　　　　BGEcol

**Figure 1.** Changes in the appearance of conventional (G) and organic (GEcol) garlic bulbs during the fermentation process (temperature 70 °C, 80% humidity, 45 days). GR—raw conventional garlic; GREcol—raw organic garlic; BG—conventional black garlic; BGEcol—organic black garlic.

**Table 1.** Physicochemical properties of raw and black garlic from conventional and organic cultivation.

| Sample | GR | GREcol | BG | BGEcol |
|---|---|---|---|---|
| Moisture (%) | 67.87 ± 0.99 [c] | 61.79 ± 1.06 [b] | 47.59 ± 0.92 [a] | 45.54 ± 1.96 [a] |
| Water activity | 0.9864 ± 0.004 [d] | 0.9642 ± 0.001 [c] | 0.9173 ± 0.001 [b] | 0.8919 ± 0.001 [a] |
| pH | 6.31 ± 0.03 [c] | 6.52 ± 0.04 [d] | 4.15 ± 0.01 [b] | 4.01 ± 0.01 [a] |

The measurements ware repeated three times, and all values are means ± standard deviation. [a–d]—means followed by different letters in the same line are significantly different ($p < 0.05$); GR—raw conventional garlic; GREcol—raw organic garlic; BG—conventional black garlic; BGEcol—organic black garlic.

As can be seen, the tested garlic samples differed significantly ($p < 0.05$) in terms of humidity, water activity and pH. The average water content in fresh garlic was 64.83 ± 3.45%, which corresponds

to the results of other authors: 66.1–64.21% [36,37,42]. Raw garlic from organic cultivation was characterized by significantly lower ($p < 0.05$) humidity (by about 6%), and thus a higher dry matter content than garlic from a conventional cultivation system. The fermentation processes used in the case of both conventional (BG) and organic (BGEcol) garlic caused a significant ($p < 0.05$) decrease in the water content in the product (by 20.28% and 16.25%, respectively), with no statistically significant differences in GR and GREcol humidity, which amounted on average to 46.57 ± 1.77%. The obtained results are in line with the results of other authors who showed the water content in black garlic at the level from 29.88% to 46.06% [36,37,42]. The moisture content is an important parameter influencing the consistency and texture of black garlic. As reported in the literature, the optimal moisture content ensuring appropriate softness, elasticity, and firmness of black garlic is 40–50%. The loss of water content up to 35–40% makes the product less flexible, and the decrease in humidity below 35% results in the product being too dry, hard, inelastic, and unfit for consumption [37,42]. In the process of producing black garlic, a decrease in water content is observed, which depends on the conditions of the fermentation process, such as humidity, temperature, and heat treatment time, with a lower moisture content at higher temperatures [36]. It is true that there are studies on the impact of the garlic fermentation process on its humidity [37,42], but their comparison is very difficult due to the too large diversity of key conditions of this process, i.e., temperature, humidity, and heat treatment time.

The water activity of raw garlic was on average 0.97 ± 0.01, which is in line with the results of other authors (0.97–0.98) [36,43], while the $a_w$ of GR was significantly higher than that of GREcol. The garlic fermentation process decreased the $a_w$ parameter by an average of about 0.07, while BGEcol was characterized by the lowest value of this parameter among all the subjects. The results of our own research are similar to the results of other authors who showed the $a_w$ of black garlic on the level from 0.91 to 0.98 [43]. In various studies, garlic differed significantly in the level of water activity, depending on the fermentation time and conditions (temperature and humidity) [36,43]. Nevertheless, taking into account the results of other authors and the results of our own research, it can be concluded that the decrease in $a_w$ during the aging of garlic is relatively small, which is mainly related to high humidity present during heat treatment processes [36], and the obtained values of $a_w$ do not guarantee microbiological safety of the product thus obtained and cannot be used as an indicator of this durability [43]. Therefore, it is recommended to dry black garlic to a value of $a_w < 0.85$ and seal it in barrier packaging.

Black garlic is one of the foods with a relatively long shelf life due to the low pH resulting from the fermentation process. As can be seen from the data in Table 1, the tested garlic samples were characterized by a significant ($p < 0.05$) differentiation in pH, which was on average 6.42 ± 0.11 for fresh garlic, and over 36% lower for fermented black garlic, amounting to an average of 4.08 ± 0.08. The obtained research results are similar to the results of other authors who showed that the pH of fresh garlic (6.13) drops due to the applied heat treatment (55 °C temperature, 80% humidity during 90 days) to approx. 4.0 [44]. Toledano-Medina et al. [43] showed that with the increase of the heat treatment time (90% humidity during 33 days), the pH decreased from 5.93 ± 0.01 for fresh garlic heads to 3.49 ± 0.06 for fermented garlic heads and from 6.31 ± 0.07 for fresh garlic cloves to 3.52 ± 0.03 for fermented garlic cloves, depending on the temperature. They also found statistically significant differences in the pH of fermented garlic depending on the temperature used in the aging process. Higher temperature resulted in lower pH. Other authors also found a significant decrease in the pH of raw garlic (6.42) heat treated at various temperatures (45 and 85 °C) for 45 days (5.00 and 3.05, respectively), suggesting that the decrease in pH is closely related to the processing temperature [36]. These authors also showed that heat treatment at 40 °C for 5 days causes pH to drop to 5.94, and at 80 °C to 4.09. These and other studies suggest that changes in garlic's pH depend on both the time and temperature of the fermentation process [36,40,43,44,64,65].

The lowering of the pH in fermented garlic compared to raw garlic is closely related to the browning processes. As the content of Maillard reaction products increases, the synthesis of organic acids increases as a result of the oxidation of aldehyde groups in aldoses, which causes a decrease in pH [35,44,66–68].

Additionally, Liang et al. [65] reported that the content of acetic acid increases significantly due to the degradation of hexose during the fermentation processes of garlic. Low pH and high temperature of heat treatment are the most important factors determining the microbiological safety of garlic, because pH below 4.2 allows for a significant reduction in the growth of microorganisms that cause food poisoning. Temperature above 60 °C and pH below 4.2 allow to inhibit the proliferation of anaerobic bacteria [36]. In addition, at the high temperature of the heat treatment process, low pH ensures greater effectiveness in getting rid of spores of bacteria or fungi responsible for food spoilage [69].

The available literature lacks comparative data on the pH of fermented garlic from conventional and organic cultivation. The results of our own research (Table 1) show that organic garlic, both fresh and fermented, was characterized by a significantly lower pH in relation to garlic from conventional cultivation (by 0.20 in fresh garlic and 0.14 in black garlic, respectively), which could suggest a better technological suitability of organic garlic for fermentation processes and better "preservation" effects of black garlic. For this purpose, however, further research is required, due to the need to standardize the methods of producing black garlic, because both pH and $a^w$ reach different values depending on the temperature, humidity, and time of heat treatment.

Content of total polyphenols, selected phenolic compounds (phenolic acids and flavonoids) and antioxidant activity of fresh and black garlic from conventional and organic cultivation are shown in Table 2. The tested garlic samples showed a significant ($p < 0.05$) differentiation in the content of total polyphenols, phenolic acids, and flavonoids, as well as in the antioxidant activity. The average content of total polyphenols in fresh garlic was 6.45 ± 0.87 mg GAE/g d.m., and the applied heat treatment processes caused a significant ($p < 0.05$) increase in the content of these components in black garlic (on average 13.80 ± 1.36 mg GAE/g d.m.). The results of our own research are similar to the results of other authors who obtained a significant increase in the content of polyphenolic compounds under the influence of the aging processes of garlic [40,41,43,45,70]. Toledo-Medina et al. [43] showed approximately a threefold increase in the total content of total polyphenols in fermented (at different times and at different temperatures) heads of black garlic (3.86–16.32 mg GAE/g d.m.) as compared with fresh garlic (5.13 mg GAE/g d.m.), and in fermented (under the same conditions) cloves of garlic up to a sixfold increase in the content of these bioactive compounds, (3.11–18.77 mg GAE/g d.m.) as compared with fresh garlic cloves (3.20 mg GAE/g d.m.). Similar changes were previously reported by Kim et al. [45], who noted a 4–10-fold increase in the content of total polyphenols at different temperatures, humidity levels and heat treatment time. In this study, on average, more than a twofold increase in the content of total polyphenols in black garlic was observed. The differences in the content of these bioactive compounds in comparison with the studies of other authors certainly result from the conditions of the applied heat treatment (time, temperature, and humidity). As reported in the literature, the content of total polyphenols tested as a function of time changes very dynamically, i.e., it increases intensively until the 21st day of heat treatment, and then it may remain constant or still increase, but to a lesser extent [40]. The increase in the total content of free phenolic compounds during heat treatment is probably due to their easier release from the bound (esterified and glycosylated) forms under the influence of high temperature [45,71,72]. The further increase in the polyphenol content is most likely due to the constantly increasing levels of complex phenolic compounds formed as a result of the later phases of the browning reaction [73].

**Table 2.** Content of total polyphenols, selected phenolic compounds (phenolic acids and flavonoids) and antioxidant activity of fresh and black garlic from conventional and organic cultivation.

| Bioactive Compounds | GR | GREcol | BG | BGEcol |
|---|---|---|---|---|
| Antioxidant activity µmol TEAC/g d.m. | 273.65 ± 1.86 [a] | 363.42 ± 1.44 [b] | 546.26 ± 4.60 [c] | 654.93 ± 4.54 [d] |
| Total polyphenol content mg GAE/g d.m. | 5.62 ± 0.05 [a] | 7.28 ± 0.04 [b] | 12.50 ± 0.13 [c] | 15.10 ± 0.14 [d] |
| Total phenolic acids mg/100 g d.m. | 38.67 ± 0.14 [a] | 49.91 ± 0.05 [b] | 54.97 ± 0.11 [c] | 75.50 ± 0.57 [d] |
| Gallic acid mg/100 g d.m. | 9.54 ± 0.05 [a] | 10.82 ± 0.01 [b] | 11.49 ± 0.03 [c] | 14.42 ± 0.30 [d] |
| Chlorogenic acid mg/100 g d.m. | 5.66 ± 0.01 [a] | 5.25 ± 0.05 [b] | 14.88 ± 0.02 [c] | 21.44 ± 0.37 [d] |
| Caffeic acid mg/100 g d.m. | 4.19 ± 0.03 [a] | 7.85 ± 0.07 [c] | 6.48 ± 0.01 [b] | 14.42 ± 0.07 [d] |
| P-coumaric acid mg/100 g d.m. | 18.71 ± 0.11 [a] | 25.36 ± 0.02 [c] | 21.59 ± 0.18 [b] | 24.69 ± 0.03 [d] |
| Ferulic acid mg/100 g d.m. | 0.57 ± 0.03 [a] | 0.64 ± 0,02 [b] | 0.54 ± 0.03 [a] | 0.53 ± 0.02 [a] |
| Total flavonoids mg/100 g d.m. | 35.02 ± 0.06 [a] | 49.29 ± 0.04 [b] | 53.53 ± 0.10 [c] | 63.32 ± 0.55 [d] |
| Catechin mg/100 g d.m. | 9.54 ± 0.03 [a] | 17.42 ± 0.01 [d] | 9.74 ± 0.02 [c] | 8.11 ± 0.01 [b] |
| Epicatechin mg/100 g d.m. | 3.17 ± 0.01 [a] | 3.73 ± 0.01 [c] | 3.29 ± 0.02 [b] | 3.94 ± 0.04 [d] |
| Gallate epigallocatechin mg/100 g d.m. | 4.46 ± 0.01 [c] | 6.38 ± 0.01 [d] | 3.72 ± 0.03 [b] | 2.94 ± 0.05 [a] |
| Myricetin mg/100 g d.m. | 3.13 ± 0.00 [a] | 3.37 ± 0.02 [b] | 3.46 ± 0.02 [d] | 3.43 ± 0.01 [c] |
| Quercetin mg/100 g d.m. | 11.11 ± 0.01 [a] | 10.89 ± 0.01 [a] | 26.06 ± 0.03 [b] | 34.39 ± 0.51 [c] |
| Kaempferol mg/100 g d.m. | 3.62 ± 0.02 [a] | 7.50 ± 0.02 [c] | 7.26 ± 0.02 [b] | 10.51 ± 0.02 [d] |

The measurements ware repeated three (flavonoids) or six times (antioxidant activity and total polyphenol content), and all values are means ± standard deviation. [a–d]—means followed by different letters in the same line are significantly different ($p < 0.05$). GR—raw conventional garlic; GREcol—raw organic garlic; BG—conventional black garlic; BGEcol—organic black garlic.

In the available literature there are insufficient comparative data on the effect of cultivation conditions in the conventional and ecological systems on the content of bioactive components (total polyphenols, phenolic acids, or flavonoids) and on the antioxidant activity in fermented black garlic. According to our research, the total content of total polyphenols in fresh organic garlic (GREcol) was on average 7.28 ± 0.04 mg GAE/g d.m. and was significantly higher (almost 23%) than in conventional garlic (GR). The applied heat treatment of garlic contributed to a significant increase in the content of these bioactive compounds (2.2 times for BG and 2.1 times for BGEcol), making the latter the product with the highest total of the total polyphenols (15.10 ± 0.14 mg GAE/g d.m.).

The tested garlic was characterized by a high content of phenolic acids (on average 44.29 ± 6.16 mg/100 g d.m.), which increased significantly during the applied heat treatment (65.23 ± 11.25 mg/100 d.m.). As in the case of the total polyphenols, the sum of phenolic acids in GREcol (49.91 ± 0.05 mg/100 g d.m.) was significantly ($p < 0.05$) higher (by about 26%) than in GR. Under the influence of heat treatment, both in conventional (BG) and organic (BGEcol) garlic,

the content of phenolic acids increased by approximately 42% and 51%, achieving the highest content of these components in BGEcol (75.50 ± 0.57 mg/100 g d.m.). The existing research by other authors showed a similar tendency of changes in the content of these compounds under the influence of the aging processes of garlic [36,43]. Kim et al. [45] showed an approximate 4.6–7.8-fold increase in the content of phenolic acids as a result of the applied heat treatment, while concluding that the main phenolic acids in black garlic at different stages of heat treatment were derivatives of hydroxycinnamic acid, which has also been confirmed in our own research. Both in BG and BGEcol garlic, cinnamic acid derivatives constituted about 80% of the determined phenolic acids, while in BGEcol the content of hydroxycinnamic acids was 61.08 mg/100 g and was approx. 17.59 mg/100 g d.m. higher than in BG (43.49 mg/100 g). The increase in the content of phenolic acids during the heat treatment of garlic is attributed to their high stability and easier release as compared with other substances with a phenolic structure [45,74–77]. According to other authors, the differentiation in the share of individual phenolic acids at different stages of heat treatment most likely results from different mechanisms of their synthesis and metabolism during fermentation processes [45].

The tested garlic samples were also characterized by a differentiated ($p < 0.05$) content of flavonoids. The average content of these compounds in fresh garlic was 42.16 ± 7.82 mg/100 g d.m., while the content of flavonoids in fresh organic garlic was significantly higher (49.29 ± 0.04 mg/100 g d.m.) than in conventional garlic (35.02 ± 0.06 mg/100 g d.m.). Under the influence of garlic fermentation processes, the content of flavonoids increased, reaching an average of 58.42 ± 5.38 mg/100 g d.m. in black garlic, with the highest content of these bioactive compounds again recorded for BGEcol (63.32 ± 0.55 mg/100 g d.m.). The results for black garlic obtained in this study are similar to the data in the literature [45].

The increasing tendency of the content of flavonoids observed in this study under the influence of heat treatment is reflected in the literature [71,77,78]. Kim et al. [45] recorded 1.3–3.5 times increase, and Choi et al. [40] 2–5 times increase in the content of flavonoids in black garlic, depending on the time and conditions of fermentation. However, due to the significant differentiation of the conditions of the applied heat treatment, the comparison of the share of particular groups of flavonoids or specific compounds is difficult and unreliable. Flavonoids are a very diverse group of phenolic compounds, differing not only in their chemical structure (a different number and distribution of hydroxyl groups), but also in thermal stability [74]. For example, in our research, a relatively high content of kaempferol was found in both fresh (5.56 mg/100 g d.m. on average) and fermented garlic (8.89 mg/100 g d.m. on average), while Kim et al. [45] did not detect it at all. These authors also did not show the presence of quercetin in fresh garlic, but only in black one, and in this study it was found on average 11.0 ± 0.12 mg/100 g d.m. in fresh and almost 3 times more (30.22 ± 4.58 mg/100 g d.m.) in black fermented garlic. As can be seen from the data in Table 2, among the tested flavonoids, flavanols had the greatest share both in fresh and fermented garlic. GREcol was characterized by a significantly higher content of quercetin and kaempferol, and a significantly lower content of myricetin as compared with BG. No such unambiguous regularity was found with regard to the selected catechins identified in the study.

Considering potential health benefits, black garlic's most salient property is its high antioxidant potential. As can be seen in Table 2, the tested garlic samples were characterized by a strong ability to inactivate the synthetic ABTS$^{+\bullet}$ cation radicals, and thus by a high antioxidant activity. Statistically significant ($p < 0.05$) differences were found between all the types of tasted garlic samples. The average antioxidant activity of fresh garlic (all fresh garlic samples, i.e., conventional (GR) and fresh organic (GREcol) garlic) was 318.54 ± 46.91 µmol TEAC/g d.m. The applied heat treatment processes increased the antioxidant activity of black garlic to an average of 601.60 ± 55.88 µmol TEAC/g d.m. The significantly highest antioxidant potential among all the tested samples was in organic black garlic (BGEcol) and reached the value of 654.93 ± 4.54 µmol TEAC/g d.m.

Raw materials from organic cultivation are characterized by a higher content of phenolic bioactive ingredients, influencing their higher antioxidant potential, which was confirmed in our own research.

Higher concentration of polyphenols compounds in organic black garlic bulbs can be explain by C/N theory [79]. Plants in organic agriculture with limiting organic nitrogen fertilization in first time of their metabolism start to produce compounds with carbon (C.). Polyphenols belong to this group secondary metabolites without nitrogen in their structure [80]. In conventional agriculture when mineral nitrogen fertilization is widely used, plants in first moment start to produce compounds with nitrogen (amino acids, peptides, proteins as well N-containing secondary metabolites, such alkaloids). On the other hands in organic farming using of synthetic plant protector are completely forbidden. Plants have to use they own protection systems such higher concentration of phenolic compounds. Higher concentration of polyphenols are an effect of biotic stresses as pest and diseases [81].

The results of the antioxidant activity obtained in this study are similar to the literature data [40,82,83] and confirm the increase in the antioxidant properties of garlic under the influence of the fermentation processes used (twice in this study), while the available literature did not show any data on the influence of the fermentation process on the antioxidant activity of conventional and organic black garlic.

Toledo-Medina et al. [43] noted a 6.5-fold increase in the antioxidant activity of fermented black garlic cloves and a 9.5-fold increase in this parameter for fermented garlic heads. Additionally, they showed that the antioxidant activity of garlic increases with the increasing process temperature, peaking at 72 °C. A similar relationship was also shown by Bae et al. [36]. During the heat treatment of black garlic at 80 °C, they noted a more than sevenfold increase in the DPPH radical scavenging ability, and at 40 °C—by half as much (3.6-fold), which would confirm the positive relationship between the antioxidant properties of black garlic and heat treatment temperature. Other authors, examining the changes in the antioxidant activity of black garlic over time, have shown that it increases intensively (about twofold) until the 21st day of fermentation, and then it slightly decreases [40].

The basic substances with antioxidant properties in black garlic, whose content increases many times as a result of fermentation processes, include polyphenol compounds, sulfur compounds being alliin and allicin derivatives, i.e., diallyl sulfides, disulfides and trisulfides, and S-allyl-cysteine, whose content may increase even six times, which results in even several dozen times higher antioxidant activity of black garlic [37,41]. S-allyl-cysteine is produced by the enzymatic hydrolysis of γ-glutamyl-S-allylcysteine (GSAC) present in significant amounts in raw garlic (GSAC) under the influence of the temperature-dependent γ-glutamyl transpeptidase (γ-GTP) [32,40,84,85]. The activity of γ-GTP is also enhanced by the high water content in fermented garlic [36,38,50]; therefore, the fermentation conditions present during the aging of garlic, mainly temperature and humidity, determine the amount and share of individual organosulfur metabolites to a wide extent, and thus also the antioxidant properties of the final product obtained. The increase in the total antioxidant activity in black garlic is explained in the literature not only by an increase in the content of total polyphenols or S-allyl-cysteine [82], but also by the content of browning products formed in Maillard reactions during the aging of garlic, also responsible for the color of black garlic [46,47,86–88].

Similarities and differences found in the tested quality parameters in tested garlic samples are shown in Figure 2. This projection shows the distribution of the results on the plane formed by the selected two factors which were responsible for the variability of the samples in 86.80% (factor p1 and p2). In other words, the projection is a correlation image of the obtained data. The location of the garlic samples in a coordinate system built using two main components shows that they were characterized by large diversity of evaluated parameters. It was found that the tested garlic samples were divided into three groups: the BG and BGEcol samples (first group), the GR sample (second group) and the GREcol sample (third group), which means that they were characterized by a diversity of the evaluated parameters (in the projection, the sample groups were located far away from each other). In the case of BG and BGEcol, the results were very similar in respect of total flavonoids, total phenolic acids, total polyphenol content, quercetin, kaempferol, chlorogenic acid and caffeic acid. These samples were different from the GREcol and GR samples in terms of p-coumaric acid, catechin, gallate epigallocatechin and moisture content. At the same time, the GR sample had a higher content

of gallic acid, ferulic acid, myricetin, epicatechin and higher aw and pH values than BG, BGEcol and GREcol samples.

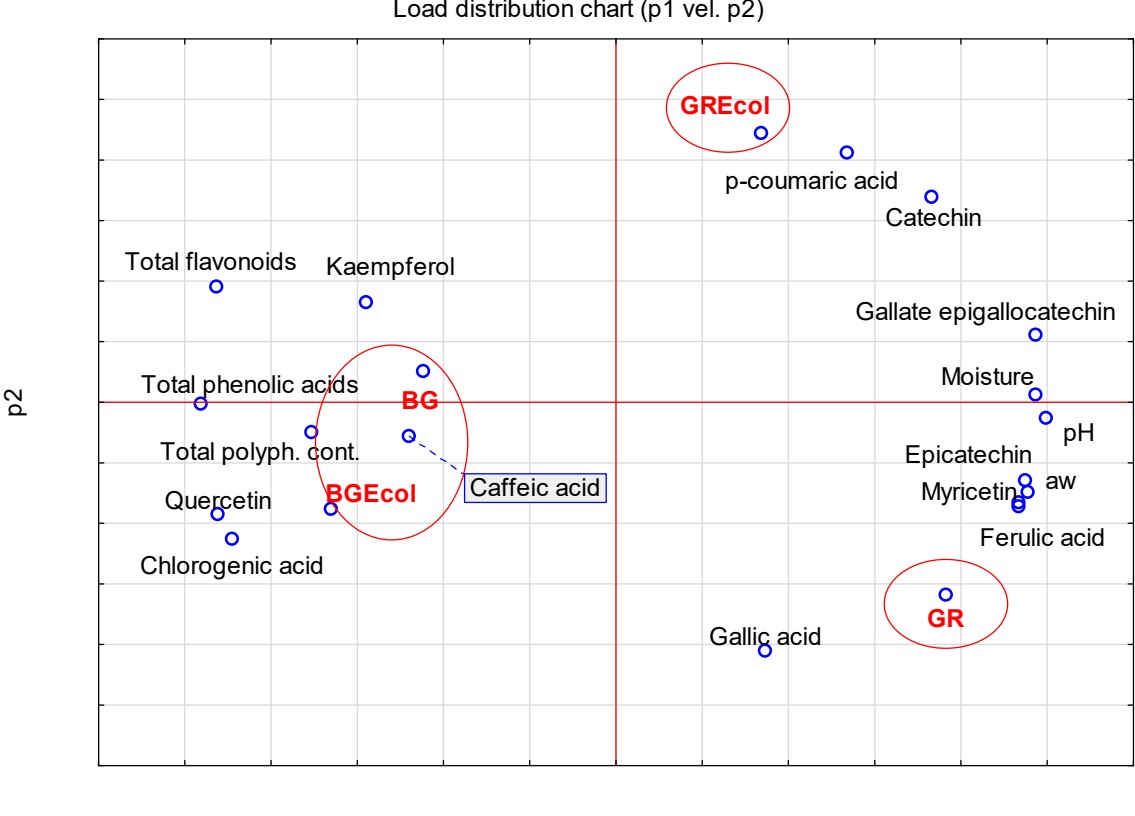

**Figure 2.** Similarities and differences found in the tested quality parameters in tested garlic samples.

## 4. Conclusions

Heat treatment processes are usually associated with the degradation of various chemical compounds naturally occurring in fruit or vegetable raw materials, and thus with the loss of nutritious bioactive ingredients. Numerous studies confirm the negative impact of heat treatment (e.g., frying, baking, boiling, microwave, or convection drying) on the content of polyphenolic compounds, natural flavors, and dyes (e.g., chlorophylls or carotenoids), vitamins or antioxidants, and thus reducing the health-promoting properties of products and their nutritional value.

Long-term heat treatment of fresh garlic under high temperature and humidity, called fermentation or aging of garlic, produces black garlic, a product with completely different physicochemical, organoleptic and sensory properties compared to fresh one. As a result of the transformation of carbohydrates (Maillard reactions), phenolic and organosulfuric compounds taking place during it, not only the color of garlic changes (to dark blue-black), but also the aroma (complete lack of the typical garlic aroma), taste, texture (soft, velvety, and delicate taste), and most importantly, the content of bioactive ingredients (total polyphenols, flavonoids, and phenolic acids), and antioxidant activity are increased.

Our research shows a significant increase in the content of total polyphenols (12.50 ± 0.13 mg GAE/g d.m.), flavonoids (53.53 ± 0.10 mg/100 g d.m.), phenolic acids (54.97 ± 0.11 mg/100 g d.m.), and an increase in antioxidant activity (546.26 ± 4.60 μmol TEAC/g d.m.) in black garlic compared to fresh one (5.62 ± 0.05 mg GAE/g d.m., 35.02 ± 0.06 mg/100 g d.m., 38.67 ± 0.14 mg/100 g d.m., and 273.65 ± 1.86 μmol TEAC/g d.m., respectively). Additionally, fermented black garlic from organic cultivation is characterized by a significantly higher content of bioactive ingredients (by about 20% for

total polyphenols, 28% for flavonoids, and 37% for phenolic acids content) and a higher antioxidant potential (by about 80%) than black garlic from conventional cultivation. Therefore, the results of our research can be a valuable hint for consumers, whose nutritional awareness, interest in organic food, and care for health and the quality of meals are constantly growing. The modern food and processing industry must meet not only the requirements as to the quality and health standards of food products, but also meet the high demands of the consumers looking for food with high nutritional, dietary and health-promoting values. Therefore, comparative research on physicochemical, technological, bioactive, and health-promoting properties is important for many industries and the food market.

Considering that the physicochemical and bioactive properties of black garlic, including the composition and mutual proportions of biologically active substances, depend on the origin, variety, cultivation method and, above all, on the conditions of heat treatment of the raw material (in particular temperature, humidity and fermentation time) further research is necessary to select the most optimal technological processes and conditions of their course, in order to maintain or obtain products with the best pro-health, dietary and nutritional values.

**Author Contributions:** Conceptualization, K.N.; methodology, K.N., A.S. and E.H.; software, K.N.; validation, K.N., A.S. and E.H.; formal analysis, K.N.; investigation, K.N.; resources, K.N.; data curation, K.N.; writing—original draft preparation, K.N.; writing—review and editing, K.N.; visualization, K.N.; supervision, K.N.; project administration, K.N.; funding acquisition, K.N. All authors have read and agreed to the published version of the manuscript.

**Funding:** This research received no external funding.

**Acknowledgments:** This paper has been published under the support of: Institute of Human Nutrition Sciences, Warsaw University of Life Sciences (WULS).

**Conflicts of Interest:** All authors declare that they have no conflict of interest.

**Human and Animal Rights Statement:** This article does not contain any studies with human or animal subjects.

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
