# Peer review of "Influence of Thermal Processing on the Bioactive, Antioxidant, and Physicochemical Properties of Conventional and Organic Agriculture Black Garlic (Allium sativum L.)"

_applsci, doi:10.3390/app10238638_

Round 1

Reviewer 1 Report

My correction with the manuscript are relatively minor, and are in the attached pdf file. I can recommend publication of this paper after these corrections.

Author Response

Dear Reviewer.

Thank You very much for Your favorable review. Thank You also for relevant and valuable comments. I have made corrections according to Your suggestions. I marked all changes in yellow. I explained the abbreviations, corrected the misspelled abbreviations, replenished the reagent sources, all of them were from Sigma-Aldrich. I put the parentheses wherever You suggested. I also changed the way of expression in all the places You indicated (as suggested). I changed the units in table 2. I deleted the duplicate phrases in the text. I clarified the statement regarding the value of the antioxidant property, I mean the average antioxidant activity of all fresh garlic samples, both conventional and organic, compared to black garlic (also for all samples, both conventional and organic). Perhaps it was vaguely spelled, so I changed it to make it understandable. I hope that's what the Reviewer meant. I put a caption under graph 2 (instead of above the graph). I also clarified the conclusions. In the list of references, I have written latin names in italics wherever it was needed. I only have doubts about the letters that indicate statistical differences. Honestly, I have not heard of a record that letters are placed next to averages, usually they are placed after the full notation, mean ± standard deviation. For now, I have allowed myself to leave the original note, but if You find it necessary, I will of course change it.

Thank You again for a very favorable review

Best regards

Katarzyna Najman

Reviewer 2 Report

In the article titled "Influence of fermentation processes on the bioactive, antioxidant and physicochemical properties of conventional and organic agriculture garlic (Allium sativum L.)" the effect of the growing method on the content of bioactive compounds in black garlic is analysed.

The authors described the garlic properties and usage, especially the health effects and so-called fermentation of garlic. The advantage of the presented study is the examination of the impact of the processing process on raw materials grown using conventional or ecological methods.

General opinion

The text is well written and almost ready for publication. But I have a few suggestions to improve the readability of the manuscript.

As the authors mentioned: "process takes place without the participation of microorganisms (bacteria, fungi)" [line 107]. I strongly recommend not to use the word "fermentation" regarding garlic processing, at least in the title. It is misleading the reader who does not have sound knowledge about the ageing of garlic.

I propose to shorten the part of the Introduction [lines 45-81] which provide a lot of details related to health effects and compounds of garlic, in my opinion not necessary in the text related to food processing. Please, consider the most important characteristic related the carried out research.

Lines 151-160 In my opinion, a better section to such consideration is the Results or Conclusion.

Author Response

Dear Reviewer.

Thank You very much for the favorable revier. Thank You also for relevant and valuable comments. I applied a correction according to Your suggestions. I changed the title to " Influence of thermal processing on the bioactive, antioxidant and physicochemical properties of conventional and organic agriculture black garlic (Allium sativum L.)". I greatly shortened the introduction in lines 45-81 (as suggested). I have also moved the part indicated by You in lines 151-160 to the summary. I think it will be a better place.

Thank You again for a very favorable review

Best regards

Katarzyna Najman

Reviewer 3 Report

The paper is very interesting, and well written but, in my opinion, few points should be improved.

  • Paragraph 2.1: Indicate the year of collection of studied sample. Was a part of the sample kept and deposited to be available for future studies?
  • Change "ml" with "mL"
  • Uniform decimal digits in the text (always 2 or 3 decimal digits).

Author Response

Dear Reviewer.

Thank You very much for Your favorable review. Thank You also for relevant and valuable comments. Following Your suggestion, I've tweaked "ml" to "mL" throughout. I also unified the decimal values ​​in the text (to two decimal places). I only left the values ​​for water activity up to the fourth decimal place in table 1 and the standard deviation up to the third decimal place also for water activity. Elsewhere the values ​​are written to the second decimal place. Regarding point 2.1, as You suggested, I completed the sampling year (2019).  All tests were performed in 2019. Regarding the deposit of the material for further research. Unfortunately, a small part of the sample remains for further research (it has been frozen), because the vast majority has already been used for physicochemical determinations for this article, but also for physicochemical and sensory determinations for the next article, which I am currently working on and I would like to publish it soon as a continuation of this article. research.

Thank You again for a very favorable review

Best regards

Katarzyna Najman

Reviewer 4 Report

This paper report s a detailed comparison of variances in Phenolic and flavonoid-like metabolites (alongside pH and global antioxidant activity) between raw and fermented black garlic preparations from garlic bulbs cultivated under both conventional and organic practices.

The quality of the data is good and it is thoroughly analysed and critiqued within the context of less comprehensive, but related analyses reported b others.

Overall the paper makes a valuable and informative contribution to the research field and merits publication in   the Applied Sciences Journal, subject to some minor corrections listed below.

1. It would greatly enhance the paper if the authors were able (if possible) to provide/propose some explanations for the significantly elevated levels of the phenolic/flavonoid/antioxidant levels observed in the fermented garlic from organically farmed compared to conventionally farmed sources. what difference in the conventional and organic farmed practices could potentially cause this.

2. line-72:  .....change glutaouryl to glutamyl (please correct elsewhere throughout the text|).

Author Response

Dear Reviewer.

Thank You very much for Your favorable review. Thank You also for relevant and valuable comments. I have made corrections according to Your suggestions. I corrected the misspelled name to the correct one, which is "glutamyl". Thank You for catching this error. I have also completed some explanations regarding the potential possibilities and reasons for the higher bioactive compounds and antioxidant activity of organically grown garlic. I put them on lines 424-435. I also added relevant references to the text (items 79-81)

Thank You again for a very favorable review

Best regards

Katarzyna Najman
